# Artificial Intelligence for Quality Defects in the Automotive Industry: A Systemic Review

**DOI:** 10.3390/s25051288

**Published:** 2025-02-20

**Authors:** Oswaldo Morales Matamoros, José Guillermo Takeo Nava, Jesús Jaime Moreno Escobar, Blanca Alhely Ceballos Chávez

**Affiliations:** 1Centro de Investigación en Computación, Instituto Politécnico Nacional, Ciudad de México 07700, Mexico; omoralesm@ipn.mx; 2Escuela Superior de Ingeniería Mecánica y Eléctrica, Instituto Politécnico Nacional, Unidad Zacatenco, Ciudad de México 07738, Mexico; jtakeon2400@alumno.ipn.mx (J.G.T.N.); bceballosc0800@alumno.ipn.mx (B.A.C.C.)

**Keywords:** artificial intelligence, automotive industry, data, defect, fault, Industry 4.0/5.0, machine learning, manufacturing, model, neural networks, principal component analysis, quality, Quality 4.0, system

## Abstract

Artificial intelligence (AI) has become a revolutionary tool in the automotive sector, specifically in quality management and issue identification. This article presents a systematic review of AI implementations whose target is to enhance production processes within Industry 4.0 and 5.0. The main methods analyzed are deep learning, artificial neural networks, and principal component analysis, which improve defect detection, process automation, and predictive maintenance. The manuscript emphasizes AI’s role in live auto part tracking, decreasing dependance on manual inspections, and boosting zero-defect manufacturing strategies. The findings indicate that AI quality control tools, like convolutional neural networks for computer vision inspections, considerably strengthen fault identification precision while reducing material scrap. Furthermore, AI allows proactive maintenance by predicting machine defects before they happen. The study points out the importance of incorporating AI solutions in actual manufacturing methods to ensure consistent adaptation to Industry 5.0 requirements. Future investigations should prioritize transparent AI approaches, cyber-physical system consolidation, and AI material enhancement for sustainable production. In general terms, AI is changing quality assurance in the automotive industry, improving efficiency, consistency, and long-term results.

## 1. Introduction

The World Economic Forum in 2024 stated that the automotive industry can be described as the globalized group of vehicle assemblers, component and equipment suppliers, service providers, and dedicated retailers specialized in their sale and distribution [1].

The Logistics World in 2023 mentioned that the automotive sector is critical internationally and is under continuous change and development. It is an industry that generates millions of jobs and impacts the gross domestic product of many economies. It requires high capital and investments in production, marketing, and research because it is a sector driven by new technologies that require innovation and competition. Companies in this sector have a global presence, so it is necessary for them to adapt to different regulations and consumer consumption habits, and, as a result, the sector faces several challenges and opportunities, such as pressures to reduce emissions and improve fuel efficiency, competition with new players that disturb the traditional order of the industry, and changes in consumer needs. In addition, the automotive sector faces opportunities such as the growth of emerging markets (e.g., China and India) and the increasing demand for new technologies and sustainable mobility solutions [2].

The automotive industry has negative effects due to its large emissions of greenhouse gases, contamination, and waste, affecting the performance of environmental sustainability. Innovation in solutions for unsustainable automotive raw material is still lacking. During the processing and purchase of these materials, a considerable amount of material scrap is produced [3]. Since the global economy is threatened by climate change, the road to decarbonization must be sped up in the coming years. Therefore, changing international markets to reduced-carbon-energy materials, mainly from eco-friendly sources, is critical to preventing the terrible effects of global warming [4].

Mobility and transportation play a critical role in human life, and their progress has simplified many of our tasks. Car accidents have increased dramatically as vehicle usage has also increased. The World Health Organization states that 1.3 million citizens die from car collisions every year, while others face damages or handicaps. Relevant data, such as engine temperature, vibrations, and driver alcohol consumption, are stored locally and collected using the Internet of Things. To determine the root cause of an accident, the information collected about the crash is needed. The stored data will allow companies to monitor vehicles in real time, review accidents, and investigate the insurance request from the service supplier [5].

The international environmental crisis and the negative impacts that have been produced by the increase in the footprint of C02 have drawn focus to the transition to a low-carbon-growth agenda as a key enabler of economic development. According to the International Energy Agency, the mobility sector is responsible for almost one-fourth of total worldwide carbon dioxide emissions, and as a result, it is the second highest carbon-producing industry on the planet. It is forecasted that the transportation sector will further impact the growing ratios of energy demand and carbon emissions in the coming years [6].

Due to Industry 4.0, companies have become more sophisticated in their target to maximize production, quality, and profits, as well as their aim to reduce waste, production times, and costs. With continuous technological innovation and society’s desire for maximum comfort, it is necessary to develop systems with the ability to replace humans and complete tasks independently. This was the trigger for the development of artificial intelligence for machines with the ability to make decisions and think like humans [7]. Artificial intelligence (AI), in practical terms, refers to the ability of equipment to comprehend facts, think, and make appropriate conclusions. Its goal is to create devices that can perform tasks similar to those of humans, while also learning and evolving based on past experiences [8]. The automotive industry has three areas where AI solutions can be developed: smart vehicles, aftermarket and warranty management, and intelligent design and manufacturing [8].

ISO (International Organization for Standardization) is an international confederation for domestic standards. The International Standard ISO 9000:2015, called Quality Management Systems—Fundaments and Vocabulary, explains the basic theory, pillars, and terminology to develop quality management systems and other standards. The planning of foreign standards is under the responsibility of an ISO technical committee. This text was developed by the Technical Committee ISO/TC 176, Quality Management and Quality Assurance, Subcommittee SC1, so-called concepts and terminology. This document sets a model that combines the core terms, pillars, procedures, and resources related to quality, so that companies can fulfill their objectives. It is important to note that this standard can be deployed in any enterprise, regardless of the sector, size, or business model. The terminology developed in this standard is aligned in a conceptual order illustrated by a set of graphs of the definition systems that make up the word hierarchy. This is shown in Figure 3, which was adapted from [9].

This review of the literature is divided into five sections. In Section 2, we describe the process that we followed to find the articles that we included in this manuscript. In Section 3, we propose a taxonomy to classify related works found in Section 2. In Section 4, we share our main findings regarding AI applications in the automotive industry used to solve quality issues. We finish this manuscript in Section 5 with our research opportunities, trends, and closing remarks.

## 2. Materials and Methods

In this section, we share the details on the method we used to carry out our investigation. The literature search was carried out on Scopus using the following key words: automotive defect, automotive defect artificial intelligence, automotive defect PCA, automotive fault detection and diagnosis PCA, automotive defect diagnosis identification PCA, automotive defect diagnosis identification artificial intelligence, automotive quality planning artificial intelligence, automotive quality control artificial intelligence, automotive quality assurance artificial intelligence, automotive quality improvement artificial intelligence, automotive visual quality artificial intelligence, automotive tactile quality artificial intelligence, automotive auditory quality artificial intelligence, and automotive olfactory quality artificial intelligence. After entering the corresponding key words, the next step was to reduce the search criteria by selecting only the publications of the last 5 years. Subsequently, we chose the articles that first appeared as open access, which allowed us to download the file. Finally, after skimming and scanning the sections of title, abstract, and conclusions, we decided whether the article should be considered in our literature review or not. Figure 1 illustrates the process we followed to carry out the research of the literature review.

In the following section, we will share further information about the taxonomy that was used to classify the manuscripts we found in the review of the literature.

## 3. Taxonomy

In this section, we propose a taxonomy of AI applications divided into three groups: quality management (QM), zero-defect manufacturing (ZDM), and perceived quality framework (PQF). The QM-related articles were split into four subcategories: quality planning, quality control, quality assurance, and quality improvement. ZDM journals were distributed into five subcategories: detection, prediction, repair, prevention, and others. The last category, PQF research, is separated into four subcategories: visual quality, tactile quality, auditory quality, and olfactory quality. The previous concepts are graphically described in Figure 2.

### 3.1. AI Taxonomy Based on Quality Management (QM)

Manufacturing companies aiming to improve their operational and financial results must prioritize quality management (QM) and its tools, such as quality improvement (QI), quality control (QC), and quality assurance (QA), to maintain or increase complete quality. The typical QM approaches such as lean manufacturing, theory of constraints (ToC), six sigma, total quality management (TQM), and six sigma lean have been used for many years in the industry to improve quality, products, processes, or services. Unfortunately, related work misses the holistic approach of QM to address all elements of a manufacturing system. In addition, since the approaches were created many decades ago, they have left a gap between actual market and sustainability needs. Fortunately, certain technologies immersed within Quality 4.0 and Industry 4.0 have enabled a more interconnected and effective production system [10].

QM describes the handling of the activities and actions needed to maintain distinction, considering the definition of quality policy, the formation and expansion of planning, and QA, QC, and QI, which includes practices related to planning, implementing, assuring, controlling, and improving. This framework is visualized in Figure 3 [10].

**Figure 3 sensors-25-01288-f003:**
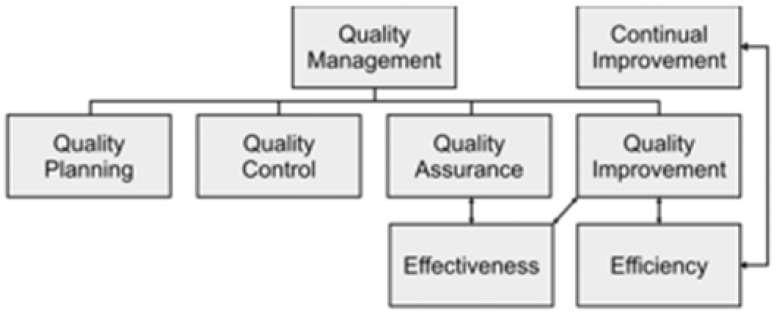
QM framework [10].

#### 3.1.1. AI Taxonomy Based on Quality Planning

To improve profitability, it is key to avoid waste and standardize production activities. By prioritizing efforts on value-added activities during the value chain, waste is minimized. Artificial intelligence (AI) can solve issues related to technical, social, and organizational impacts on production processes. Some tasks that have received benefits are sales planning, scheduling, delivery planning, capacity planning, maintenance, and quality control. Actual investigations suggest that production planning and control (PPC) topics can be addressed with machine learning (ML) tools. The method used to identify AI cases in planning was a qualitative research technique, conducting 12 interviews with machine, parts, and electronic manufacturing companies. The use cases and challenges can be summarized as follows: lack of control of orders, complexity in capacity planning, difficult alignment among demand and capacity, ineffective order authorization, and lack of prevention/detection of faults in production processes. In the literature research, it was found that forecasting of demand can be improved with AI. In addition, ML can increase prediction accuracy. Deep-reinforcement learning can be used to optimize scheduling, in order to offer savings [11].

The second journal for this category is called “Demand forecasting of spare parts with regression and machine learning methods: Application in a bus fleet”. Robust demand estimation positively impacts many aspects of a company. The projection of demand for spare parts is key to ensure that processes are not interrupted and to be ready for sales variation and other unexpected changes. For specific activities such as maintenance and repair, replacement parts must be available in the required quality, time, and quantity. Replacement parts and maintenance are critical for vehicle fleets, due to their cost being 10–60% of total expenses. In this journal, demand estimation was deployed with several regression models, a rule-based procedure, a tree-based approach, and artificial neural networks, which was the technique with the highest accuracy prediction rate and lowest deviation [12].

The third publication is “Application Scenarios of Artificial Intelligence in Electric Drives Production”. The investigation performed should narrow the gap between theory and practice by sharing specific solutions. Electric drive manufacturing has progressively become important because of big trends, such as electric mobility and process automation, which are supported by electric drives. Sets of AI tools could help the production planner select the best manufacturing processes for a certain electric drive design. In addition, those digital tools could be utilized to create intelligent systems that can adapt to changing conditions. This document presents applications of knowledge-based systems (KBS) and machine learning (ML). It was concluded that KBS can be a key player in the scheduling of electric drive manufacturing systems. Meanwhile, the ML model is based on a transcendental likelihood that maximizes individual production processes [13].

The next article is “Early Quality Prediction using Deep Learning on Time Series Sensor Data”. In production areas, issues that are only identified at the final stage of the production sequence may cause high rejection ratios, with important cost levels and scrap. To avoid this, an early quality estimate must be performed. Data-driven methods for projecting process quality on time can be developed. Machine learning (ML) algorithms with experience with sensor data can be used to estimate uncertain situations. In this work, an early quality taxonomy with a convolutional neural network (CNN) within an automotive case was investigated. The method followed was a gradient-based heat mapping for CNN to identify patterns, which were used to fulfill the quality projection. The results share adequate performance in terms of improving the classification precision and performing early quality forecasting [14].

#### 3.1.2. AI Taxonomy Based on Quality Control

Machine learning (ML) as part of artificial intelligence (AI) is being implemented in quality improvement to increase typical statistical quality control (SQC) processes. ML tools have reliability concerns, which is caused by a lack of clarity, generating a difficult to understand decision-making framework. The journal explores the implementation of faithful ML in metallic additive manufacturing (MAM) through the defined framework’s confidence in the model forecast. Within MAM, ML can be used to enhance build quality. The anticipatory ML tools that worked in the article are random forest (RF), gradient boosting trees (GBT), and fully connected neural networks (FCNN). The main findings are the following: predictive models have relevant recognition capabilities, and qualitative evaluations of ML models have reliability and applicability [15].

The second study is named “Explainable Predictive Quality Inspection using Deep Learning in Electronics Manufacturing”. The manufacturing of zero-defect goods is becoming a key competitive component for actual electronic manufacturing organizations. Method-based quality forecasts with ML trained in sensor information are employed to substitute typical inspections. The forecast of product quality, together with physical testing, is used to implement corrective actions for quality control. Surface mount technology (SMT) enables the consolidation and verification of sensor data to forecast process quality on time. In the study, a convolutional neural network is used, integrating fields of view (FOVs) and printed circuit boards (PCB). A heat mapping strategy was applied to point out the impact of PCBs on quality estimation. The estimative model shows acceptable results for the identification of deviations [16].

The next study is “Predictive model-based quality inspection using Machine Learning and Edge Cloud Computing”. Due to progressive market competition, the delivery of best-in-class products is a key enabler to ensure sustainable success for an organization. In addition, inspection processes are critical and require the use and adoption of the most recent and robust technologies. In this document, a forecast model for quality inspection in production is deployed, whose main part is a supervised ML algorithm to estimate the quality of the final product. This holistic method considers data collection, representation, model adoption, technological implementation, and IT plant infrastructure. The results indicate that inspection activities can be reduced and positive finance savings can be generated [17].

#### 3.1.3. AI Taxonomy Based on Quality Assurance

The progressive challenges linked to optimizing scheduling and planning, to improve parts, to detect faults and forecast, and to delegate monotonous activities to robots provide opportunities to discover AI properties. Quality assurance (QA) refers to the follow-up of quality features of a component to identify possible defects, an activity that is typically conducted by experts. This generates delays and interruptions; we hope that it has been supported by AI implementation through automated online condition monitoring. Some examples of technologies that help image-based issue detection are scale-invariant feature transform (SIFT) and speed-up robust features (SURF). AI plays a key role in the redefinition of the manufacturing environment, whose change process is triggered by key technologies [18].

The next journal article is “Performances of an in-line deep learning-based inspection system for surface defects of die-cast components for hybrid vehicles”. One of the main activities to improve the quality assurance of the final product is the examination of the surface, which is traditionally performed manually. Deep learning (DL) approaches can be easily adjusted to recently launched products and surface issues following sample images. DL can learn characteristics from limited data, representing complex architectures, using an automated learning procedure. The attention of this article is on the evaluation of a DL defect detection model used to produce die-cast parts. The results show a true positive rate of 98%, which means that the DL model is acceptable for the industrial application reviewed and that the results can be repeated with other parts related to surface detection [19].

The next research is “Supervision controller for real-time surface quality assurance in CNC machining using artificial intelligence”. For computer numerical control (CNC) fabrication, the exterior quality of machined parts is a key aspect to rate the results of manufacturing processes. Surface quality is difficult to estimate based on the background of machinists; therefore, it is difficult to select the appropriate machining features that will meet the technical requirements of the machined components. The effects of low surface quality negatively impact manufacturing costs and profitability. This work shares an original method for the arrangement of an intelligent supervision controller for on-time updates to fulfill the requested exterior quality of machined components. The mock-up results express that the controller considerably reduced the discrepancy among the requested and estimated surface roughness, which demonstrates that the developed model can help produce top-quality machining products [20].

#### 3.1.4. AI Taxonomy Based on Quality Improvement

The machining sector faces limitations in the availability of raw materials and the increase in their costs. In addition, sustainability policies and changes in requirements must also be considered. These factors make it necessary to look for new technologies. Artificial intelligence (AI) offers many improvements in terms of quality, efficiency, and sustainability. The machining process aims to achieve technical requirements to fulfill customer satisfaction in the shortest delivery time and at the lowest cost. Some AI techniques implemented in the field of quality are: Bayesian networks, fuzzy logic, and the adaptive neurofuzzy interference system (ANFIS) [21].

The next document is “Development and Implementation of Autonomous Quality Management System (AQMS) in an Automotive Manufacturing using Quality 4.0 Concept—A Case Study”. Industry 4.0 considers the combination of information technology, the Internet of things, artificial intelligence (AI), and other tech tools to improve smart manufacturing. Quality improvement and control are essential elements of the production sector that can be accurately handled by including the concepts of Industry 4.0 and Quality 4.0. Both organizations arrange quality management principles to improve productivity, product development, and proficiency. In addition, Quality 4.0 improves product quality, services, and customer satisfaction. IoT and Quality 4.0 were used to automate the QMS studied. Traditional quality methods such as Gage R&R and six sigma were implemented. As a result, the manufacturing rates of the analyzed machines, the process yield, and the overall machining process were improved. In addition, production/inspection costs and rejection rates decreased [22].

### 3.2. AI Taxonomy Following Zero-Defect Manufacturing (ZDM)

A new mindset for quality management (QM) is zero-defect manufacturing (ZDM), which integrates traditional quality improvement (QI) methods with modern digital tools in Industry 4.0. It handles a manufacturing system following a holistic approach rather than focusing on specific items. Following the growth of Industry 4.0 with the combination of tech tolls, ZDM has placed a special focus on learning and the business environment. The concept of ZDM was initially launched by the United States military, when the Cold War happened in 1965. With increased interest from both Industry 4.0 and digital technologies, ZDM has gained popularity within research and different businesses. In addition, ZDM is an integrated QM approach that uses several methods and digital tools to efficiently achieve best-in-class quality levels in every single element of the industry. The first big step in actual ZDM was in 2020, when the image shown in Figure 4 was created by Psarommatis et al. in [10].

This framework can collect any quality element in a production system by dividing QM into four categories: detect, predict, repair, and prevent. Detect and predict were classified as trigger tactics, and repair and prevent were classified as action procedures. To fulfill ZDM, one strategy must be integrated with one triggering strategy [10].

#### 3.2.1. AI Techniques for Detection (Physical and Virtual) Level

The first document is “Toward Zero-Defect Manufacturing with the support of Artificial Intelligence—Insights from an industrial application” [23]. Its purpose is to implement artificial intelligence (AI) in an industrial application to achieve zero-defect manufacturing (ZDM). The article explains that AI can be used as an enabler of ZDM, to support quality management and complex quality issues. Poor data quality and incorrect classification of approved products complicated the modeling and reduced its precision. The preparation time to establish a ZDM-focused architecture was identified as an important obstacle to implementing the ZDM model. Data science initiatives should be initiated immediately to confirm the accuracy of the data, the results of which should be used to continuously improve the application.The subsequent document is an approach based on artificial intelligence of things for the detection of anomalies in rotating machines [24]. The framework developed, based on artificial intelligence of things (AIoT), is validated on three rotating machines. The results confirm the strength of the proposed procedure and its capability for remote real-time monitoring and control of the machines. Integrating AI with IoT is highly positive for timely follow-up, deliberation, and preventive maintenance of rotating machines. Four classification models were tested, and good results were achieved for three rotating machines with 100% accuracy.The following study of this category is “Numerical simulation of gears for fault detection using artificial intelligence models” [25]. The article presents a method for activating AI models by constructing a model of grouped parameters in gears. Samples are generated by inserting types of failures into the model. A sample matrix is used for AI training and the classification is selected to classify failure samples. Parameters such as gear stiffness and damping are adjusted to improve classification accuracy, which ranges from 80% to 100%. It is demonstrated that AI trained with simulation data can classify measured data, compensating for the lack of real data. The proposed method overcomes the issue of inaccurate classification due to insufficient training data, showing its effectiveness in fault detection in gear systems.The next investigation is “Development and comparative evaluation of various fault detection algorithms for a drum brake using artificial neural networks and a physics-based model” [26]. The study presents an AI-based fault detection framework and a system failure model to identify the most common faults in automobile drum brakes. In summary, the study presents effective methods for detecting drum brake faults using AI, with promising results in terms of accuracy and robustness.We also included the paper “Predicting Operators Fatigue in a Human in the AI Loop for Defect Detection in Manufacturing” [27]. It studied how defect inspection can be improved using machine learning models to identify and predict operator fatigue. The results were conceptual approaches for integrating operators into AI-based inspection and a fatigue monitoring system to improve working conditions. Based on experiments with defect labeling data, most of the labeling quality decreased over time, which is linked to increased operator fatigue.The subsequent study is “Semi-supervised learning for industrial fault detection and diagnosis: A systemic review” [28]. The article addresses automation of fault detection and diagnosis (FDD) through machine learning (ML) methods. A comprehensive review of the literature on semi-supervised learning (SSL) for FDD is conducted, noting difficulties in comparing results due to variations in experiments and implementations. The review highlights key journals and methods used, with an emphasis on intrinsically semi-supervised methods. The article concludes by highlighting the importance of future research to demonstrate the unique advantages of SSL in industrial data.We add the investigation “A bi-level data-driven framework for fault-detection and diagnosis of HVAC systems” [29]. The article addresses the application of machine learning methods for fault analysis in HVAC (heating, ventilation, and air conditioning) systems, highlighting their high accuracy in detection. A novel data-based framework is proposed to improve diagnostic procedures. This framework consists of a two-level machine learning model that uses principal component analysis (PCA) to reduce the dimensionality of the dataset. A correlation analysis is conducted to identify the most influential variables. The proposed model successfully identified eight types of issues, using PCA to reduce the dimensionality of the data set and the RF algorithm to improve the detection accuracy.We also include the article “Application of sensing techniques and AI-based methods to laser welding real-time monitoring: A critical review of the recent literature” [30]. The article reviews research from the past 10 years on real-time monitoring of laser welding, focusing on detection techniques and advanced technologies based on AI. Techniques such as PCA for data processing and seam tracking with CCD cameras (charge-coupled devices) for analyzing seam characteristics in real time are mentioned. The potential of AI, especially deep learning, for processing and analyzing data in welding monitoring is highlighted. The conclusion emphasizes the importance of developing intelligent quality assessment systems as an interesting and challenging field in this area.The investigation “Real-time arc-welding defect detection and classification with principal component analysis and artificial neural networks” [31] presents an innovative system for the automatic detection and classification of arc welds. It proposes the combined use of principal components analysis (PCA) and artificial neural networks (ANN) for this purpose. The ANN enables automatic detection of welding defects. Once properly trained, the ANN can discriminate between different welding defects, such as lack of penetration or reduced gas flow. Ultimately, the combined application of PCA and ANN offers an effective solution for the automatic detection and classification of defects in arc welds.The next article is “Quantitative detection of defects based on Markov–PCA–BP algorithm using pulsed infrared thermography technology” [32], in which a Markov-PCA-BP algorithm with a predictive system was developed, combining PCA with neural network theory. A network model was used to forecast the depth and diameter of the defects. The depth and diameter of the defect were correctly identified by the Markov-PCA-BP system. The proposed method was effective, considering that the prediction fault for diameter and depth was approximately 4% to 10%.Next, we included the document “Study on PCA-SAFT imaging using leaky Rayleigh waves” [33]. This article highlights the rebuilding of high-quality images of internal or surface issues through a picture identification method that combines Rayleigh wave testing and the synthetic aperture focusing technique (SAFT). The findings reveal that the PCA-SAFT image model can decrease mechanical vibrations and image artifacts. Lateral resolution of defects is optimized, and the average defect sizing error is reduced by 48.81%. Correlation studies demonstrate that as the spread distance increases, the acoustic energy of the wave surface is reduced. The PCA algorithm can efficiently reduce system noise in Rayleigh waves with leaks.The next document is “Early and extremely early multilabel fault diagnosis in induction motors” [34], which presents an intelligent multifault diagnosis method to evaluate various fault conditions in induction motors. Using PCA and decision trees, it enables accurate diagnostics. The results show that the method is effective with complete data and capable of handling data from motors operating at different frequencies and detecting faults with precision. The combination of PCA and decision trees facilitates the identification of simultaneous faults. In summary, this intelligent fault diagnosis method is proven to be effective, robust, and capable of detecting early faults with high performance in induction motor systems.Afterwards, we considered the paper “Hydrogen leakage fault classification diagnosis based on data driven in the hydrogen supply system of fuel cell trucks” [35]. The article presents a diagnostic method for classifying hydrogen leaks on the basis of data, using a K-means algorithm. PCA is used to reduce the dimensionality of the information, and SVM (support vector machine) is carried out for diagnosis. Hydrogen leaks are simulated using a fuel cell truck model, generating 1200 failure scenarios that are classified into three levels. PCA-SVM is used to improve diagnostic accuracy, achieving 91.2%, 91.2%, and 90.3% accuracy. The SSA-PCA-SVM (sparrow search algorithm) method shows higher diagnostic accuracy compared to standard SVM and PCA-SVM. The results also indicate relationships between hydrogen leak mass, ambient temperature, and pressure.The next study is “Fault detection and classification in kinematic chains by means of PCA extraction-reduction of features from thermographic images” [36]. This work presents a methodology for correctly identifying and categorizing a broad range of defective conditions in kinematic chains. It focuses on developing non-invasive methods to diagnose these fault conditions. The reduction of the matrix by PCA enables a three-dimensional representation that facilitates the visualization of defective conditions. The suggested plan is confirmed with an organizer based on ANN, achieving an accuracy of 96.8%. The extraction and reduction of statistical features through PCA allows for clear detection and visualization of fault conditions.The article “Fuel cell diagnosis methods for embedded automotive applications” [37] focuses on the importance of ensuring the durability of fuel cells in the industrialization of technologies. The methods for each stage are detailed, including the use of temperature and pressure sensors, as well as dimension reduction algorithms such as PCA. Classifiers, divided into supervised, unsupervised, and mixed categories, are crucial for determining whether samples indicate a defective state. Three main steps are outlined for the implementation of a diagnostic algorithm in vehicles, from creating a prototype to validating it on a test bench.The research “Path Planning Optimization for Driverless Vehicle in Parallel Parking Integrating Radial Basis Function Neural Network” [38] describes a novel method for optimizing parallel parking paths for autonomous vehicles using a radial basis function neural network (RBFNN) to train Bézier curve control points. The method is validated through MATLAB (R2021a) simulations and experiments using a smart mini-car with an ultrasonic sensor. The results indicate that the optimized Bézier route meets the requirements for continuity of curvature, safety, and constraint, improving parking capabilities in tight spaces. The trained RBFNN provides accurate control points for the Bézier curve, resulting in a path with gradual curvature changes that ease steering adjustments and enhance safety. The study demonstrates that the proposed method offers better curvature continuity and optimization efficiency compared to traditional paths.The next article is “Data-driven monitoring and validation of experiments on automotive engine test beds” [39]. The article focuses on the implementation of data-based issue analysis procedures to identify variations and determine root causes with the aim of reducing costs and design time. A fault isolation technique using Fisher discriminant analysis (FDA) and contribution plot (CP) is proposed. The application of these techniques is demonstrated with results for detecting a fault in a pressure sensor. In summary, the study explores the application of data-based fault diagnosis methods, highlighting the effectiveness of the multi-modal dynamic approach and the usefulness of contribution analysis in identifying root causes.The next paper is “Application of Hidden Markov Models for Fault Detection in Automotive Engines” [40], which proposes the use of Markov models to enhance the engine design process, with the aim of reducing costs and time. During development, clear fault detection is achieved after 600 samples, with a 4.33% false alarm rate and a 24.83% missed detection rate. The study points out that traditional statistical approaches are not suitable for complex systems like engines, which is why the hidden Markov model is adopted for fault detection. The application of this model in an engine test bench demonstrates its effectiveness in fault detection.The next study is “Recent Progress and Prospective Evaluation of Fault Diagnosis Strategies for Electrified Drive Powertrains: A comprehensive review” [41], which describes a systematic review of failure types and diagnostic techniques for electrified drive powertrain systems (EPDS). It addresses different failures, along with their corresponding diagnostic methods. The need to develop new methods to address unknown failures and enhance the integration of new failure modes is emphasized, underscoring the potential of artificial intelligence and big data analysis to identify early signs of failures and improve the safety and reliability of EPDS.The last publication is “Fault diagnostics in air intake system of combustion engine using virtual sensors” [42]. This article presents a method for diagnosing faults in a gasoline engine’s air intake system that are not detected by onboard diagnostic systems. The method involves generating residuals by comparing real sensor readings (MAP and MAF) with virtual sensor outputs based on models. This technique was tested in conventional port fuel injection (PFI) and gasoline direct injection (GDI) engines. Residuals were used to detect these faults, revealing that some issues were not captured by onboard diagnostics. Future research will involve developing a hybrid engine model and creating virtual sensors based on transient states.

#### 3.2.2. AI Techniques for the Prediction Level

The first study is “How can artificial intelligence enhance car manufacturing? A Delphi study-based identification and assessment of general use cases” [43]. It highlights the enormous potential of AI to reduce operational costs in automobile manufacturing and describes general use cases of AI application in this sector. It points out the lack of a comprehensive overview on the use of AI in automobile manufacturing, with most articles focusing on details of specific use cases. The availability and quality of data are identified as important factors in the application of AI in the automotive industry, and it is acknowledged that the Delphi method has limitations in producing quantitative results that meet scientific requirements.The next article is “Online quality inspection of ultrasonic composite welding by combining AI technologies with welding process signatures” [44]. The study proposes an AI method for quality inspection in composite welding. Using artificial neural networks (ANN) and random forest (RF) models, the method predicts the failure load and the quality of the weld. The results show high levels of accuracy for both models, with an average relative error of 7.1% for ANN and 99% accuracy for RF. It is suggested that the ANN model could further improve accuracy, as well as AI’s ability to explore new functions and establish correlations between inputs and outputs.The next publication of this classification is “AI-based decision model for a quality-oriented production ramp-up” [45]. The work presents an AI-based decision model designed to configure process parameters and improve production quality using a digital shadow. The results provide tools to enhance quality by employing neural networks trained with Bayesian regularization and solving minimization problems using the SQP (sequential quadratic programming) algorithm. The quality of the regression is evaluated, and the production increases are simulated to assess the performance of the adaptive model. In conclusion, an AI-based adaptive decision model is developed that improves process quality and can simulate physical manufacturing processes.The next article is “Strength and manufacturability enhancement of a composite automotive component via an integrated finite element/artificial neural network multi-objective optimization approach” [8]. This paper addresses the upgrade of a hybrid automotive structure molded by injection, using plastic over the combination of finite element method, AI, and evolutionary investigation methods. The procedure demonstrated a reduction in product deformation and manufacturing time by 10% and 62%, respectively.The next journal article is “Intelligent systems in the automotive industry: applications and trends” [46]. This article explores the extensive use of AI, technology, and intelligent systems in the automotive industry in manufacturing, diagnostics, onboard systems, warranty analysis, and design. It highlights the integration of computational intelligence methodologies, including fuzzy logic, neural networks, and machine learning, into various automotive processes. The article also discusses the potential of AI in improving corporate knowledge management and reducing warranty costs through data mining. In addition, it identifies opportunities to apply advanced technologies in different functional areas of the automotive industry, with the aim of enhancing customer satisfaction and product quality.The next research report for this category is “Application of artificial intelligence technology in the manufacturing process and purchasing and supply management” [47]. The text highlights the growing role of AI and big data in transforming supply chains and manufacturing, focusing on how these technologies are driving concepts such as smart factories and manufacturing. The ability of AI to identify defects and issues more quickly and accurately, thereby optimizing processes, is emphasized. In summary, AI is being successfully applied in various aspects of manufacturing and supply chain management, offering significant benefits in terms of quality, efficiency, and resource management.The next investigation is “Predictive maintenance enabled by machine learning: Use cases and challenges in the automotive industry” [48]. In the automotive sector, ensuring functional safety while maintaining maintenance costs has become a challenge. A way to accomplish this is through predictive maintenance (PdM). Machine learning (ML) is potentially a PdM tool, delivering cost savings and better prediction capabilities. Among the auto parts studied in the article, the drivetrain emerged as a key user of PdM.Afterwards, we considered the research called “A parallel strategy for predicting the quality of welded joints in automotive bodies based on machine learning” [49]. This research proposes a procedure to estimate the quality of weld points using ML. PCA was also used to determine the quantity of main elements to decrease the dimensionality of the data set and ease the classification of the subsets of data. There is a strong correlation between the variables; they are autonomous and can be reviewed by the PCA algorithm.We considered the text “Systematic review on machine learning methods for manufacturing processes–Identifying artificial intelligence methods for field application” [50]. The study provides a systematic review of the applications of ML in several factory environments. In production process planning, Q-learning is used to manage large volumes of data in Internet of things environments. For quality control, decision trees and convolutional neural networks (CNNs) are used for defect classification. In predictive maintenance, both supervised and unsupervised methods are combined for fault detection and estimation of industrial equipment lifespan. In summary, the study highlights the broad spectrum of ML applications in plant processes and emphasizes the importance of its integration to improve efficiency and quality in manufacturing.We also found the study “Speech Recognition Using Deep Neural Networks: A Systematic Review” [51]. In recent decades, extensive research has been conducted related to the use of machine learning in speech processing applications, specifically speech recognition. Recently, the focus has shifted towards utilizing deep learning for these applications, yielding significantly better results. This paper reviews studies on the use of deep learning in speech processing, analyzing 174 publications from 2006 to 2018. Most of these works focus on speech recognition, with 79% of the studies concentrated in this area. The study suggests exploring new feature extraction methods and using hybrid models and recurrent neural networks (RNNs) in future research, as they offer great potential in speech recognition.In the publication “Optimal feature selection on Serial Cascaded deep learning for predictive maintenance system in automotive industry with fused optimization algorithm” [52], some data-driven predictive maintenance techniques were selected. To prevent production manufacturing shutdown, it is key to implement an efficient predictive maintenance model. Those parameters that were specified are subjected to hybrid deep learning based on Python (v. 3.8) to predict the occurrence of defects. This system´s precision and accuracy outperform other approaches.The article “Deep Learning based Predictive Testing Strategy in the Automotive Industry” [53] explains an abstract structure following deep neural networks to forecast the result of a quality control test using automobile arrangement and data processing. By applying the testing framework, the efficiency can be increased by 15%. The indications of a great method are that it achieves a comparatively high NPR (negative prediction rate) while maintaining the FOR (false emission rate) at a minimum value.The research “Adaptive-neural-network-based robust lateral motion control for autonomous vehicle at driving limits” [54] introduces a new lateral motion control method designed to enhance the stability and tracking accuracy of the trajectory of autonomous vehicles under challenging driving conditions. The proposed control system integrates a strong steering controller with an adaptive neural network (ANN) approximator. The robust steering controller aims to suppress lateral deviation, handle external disturbances, and maintain vehicle stability. The method was tested through simulations and real-world experiments, showing that it effectively maintains vehicle stability and improves trajectory tracking performance compared to traditional methods. The control scheme demonstrated superior robustness against unknown disturbances, with significant improvements in dynamic performance and trajectory tracking accuracy.Then, we looked at “Optimization of green sand process for quality improvement in castings by using combination of Taguchi Techniques-GRA-PCA” [55]. Here, components produced through sand casting have many quality features. The perspective of the publication is based on optimizing the key process specifications to reduce the presence of issues, applying Taguchi’s design of experiments (DoE) combined with gray relational analysis (GRA) and principal component analysis (PCA). According to variance analysis, the main process parameter that affects the result is the carbon equivalent (CE), with a contribution of approximately 86%.The following paper is “Life prediction of copper wire bonds in commercial devices using principal component analysis” [56], which develops a data-driven predictive model using PCA to estimate the failure time of cable joints for plastic components, which is necessary to predict the reliability of the system or module. PCA is used to detect trends and affinities within a big list of features affecting a component, reducing the number of parameters from 14 to 9. In this method, the higher the volume of data, the higher the precision that can be expected.The investigation “Prediction of Thermal Aspects for Brass Material-Based Natural Convection Heat Transfer System by Using Adaptive Neuro-fuzzy Inference System” [57] presents an adaptive neuro-fuzzy inference system (ANFIS) model created to forecast the performance of a natural thermal transfer system. The ANFIS approach effectively simulates how variations in input parameters such as current and voltage affect temperature responses at different locations in the system. Comparison with experimental data shows that the ANFIS model accurately predicts temperatures, especially at lower temperature points, such as the fourth sensor location.

#### 3.2.3. AI Techniques for the Repair Level

The first publication is “Component design optimization based on artificial intelligence in support of additive manufacturing repair and restoration: Status and future outlook for remanufacturing” [58]. The study addresses automating repair and restoration in the context of the increasing popularity of metal additive manufacturing (AM), which is still considered limited. It provides a comprehensive overview of the application of AI to optimize repair design through additive manufacturing. The study emphasizes the need to develop a systematic approach to optimize the design of components that facilitates repair through AM from the early stages of the design.The next publication of this classification is “Digital Management Systems in Manufacturing Using Industry 5.0 Technologies” [59], in which the industrial revolutions up to Industry 5.0 are discussed. The first four industrial revolutions introduced significant technological advancements. Industry 5.0 is emerging as a new era in which humans and machines will work collaboratively, using technologies such as cobots, digital twins, exoskeletons, smart materials, advanced AI, and IoT. This revolution aims to improve efficiency, customization, and environmental sustainability in manufacturing. Industry 5.0 emphasizes human–machine collaboration and the integration of advanced technologies to improve manufacturing processes and reduce environmental impacts.The subsequent article is “Artificial Intelligence and Advanced Materials in Automotive Industry: Potential applications and perspectives” [7]. A literature review of AI applications in the automotive sector is provided. It offers solutions to make automotive design, manufacturing, aftersales, and the vehicle itself more intelligent. The techniques explained make vehicles smarter, safer, and more reliable. In addition, it seeks to automate processes, reduce manual labor, improve efficiency, and eliminate repetitive tasks. AI is highly versatile and has numerous applications in the automotive sector.

#### 3.2.4. AI Techniques for the Prevention Level

The article considered within this category is called “Hierarchical decision making for proactive quality control: system development for defect reduction in automotive coating operations” [60]. This article describes that automotive coating is an example of product quality control that needs the combination of quality prediction and inspection. Furthermore, it shares a novel approach to a quality control method to handle complex issues in manufacturing systems. A case study is developed for the quality control of a vehicle topcoat application that implemented an intelligent decision-making system. This model can evaluate the execution of the process and helps prevent defects at several phases of the coating application. The effectiveness of this method relies on a tool named DRACO (Defect Reduction in Automotive Coating Operations). In conclusion, the authors share that the described methodology has relevance in many manufacturing applications.

#### 3.2.5. Others

The initial paper is “An Artificial-Intelligence-Based Method to Automatically Create Interpretable Models from Data Targeting Embedded Control Applications” [61]. The study presents an automated approach to model creation, simplifying engineering in control and calibration functions. Various potential implementations of the system are mentioned, such as method development, system recognition, organizational enhancement, and model decline. The feasibility of the method is demonstrated with two examples, highlighting its efficiency in achieving accurate results with fewer calibration parameters. Additional potential applications of the method are mentioned, including model compression, runtime optimization, system identification, and data analysis.The next document is “Artificial Intelligence (AI) and Its Applications in Indian Manufacturing: A Review” [62]. This text addresses the adoption of AI in the manufacturing industry, within the context of India. It highlights how AI is revolutionizing the industry through automation and intelligent robots, especially within the framework of Industry 4.0. In India, the manufacturing sector is controlled by small and medium enterprises (SMEs), which have difficulties in adopting advanced technologies such as AI due to lack of adequate infrastructure and specific government policies. In addition, the text addresses other topics, such as unemployment and its contributors, such as lack of relevant skills and preference for traditional manufacturing methods. The text also emphasizes the need to update the educational system so that the benefits of new technologies can be widely accessible.The study “Comparison of FDA-based and PCA-based features in fault diagnosis of automobile gearboxes” [63] presents a method to assess a multi-speed automotive gearbox. In the mentioned document, the gears analyzed are positioned on the main shaft, which is backed by a tachometer sensor. Fisher discriminant analysis (FDA) was applied to address a dimensionality issue. Failure analysis was compared with PCA, and it was found that the FDA-based diagnostic method has higher accuracy and lower monitoring costs. The failures studied were classified into three categories: problems in gear 2, problems in gear 3, and combined problems. Compared to PCA, FDA provides more acceptable results in classifying single or more complex gear failure conditions, improving average recognition by approximately 14%.The subsequent document is “Multi-criteria decision making with PCA in EDM of A2 tool steel” [64]. The effect of the process parameters lp (current), Ton (on-time), Tau (duty cycle), and V (discharge voltage) was studied using a complete factorial design methodology in the Electro Design Machine (EDM) for A2 steel, and significant effects on the selected responses MRR (material removal rate) and EWR (electrode wear rate) were considered from both qualitative (main effect plots) and quantitative (ANOVA) perspectives. The ANOVA results indicated that lp and Ton are the main process parameters. PCA was performed to find the balance between the MRR and EWR responses. Furthermore, it was found that Lp is directly aligned to MRR and EWR and has the highest input to the MRR, while Ton has the main impact on the EWR.The following journal article is “Compact Base Station Antenna Based on Image Theory for UWB/5G RTLS Embraced Smart Parking of Driverless Cars” [65]. This paper introduces a compact base station antenna designed for real-time location systems (RTLS) using ultra-wideband (UWB) and 5G technologies, aimed at enhancing smart parking solutions for autonomous vehicles. The design improves the precision of RTLS systems for autonomous vehicle parking by maintaining stable radiation patterns throughout the frequency range. Future work will involve real-world tests with autonomous vehicles.The next publication is “Hierarchical k-nearest neighbors classification and binary differential evolution for fault diagnostics of automotive bearings operating under variable conditions” [66]. This work focuses on developing a diagnostic system to detect early degradation, isolate faulty bearings, and classify the types of defects. This framework is developed according to a layered organization of K-nearest neighbors (KNN) classifiers. The overall performance of the model has been validated with validation data, demonstrating a correct classification rate of 79.78%. The C2 classifier is identified to have the least satisfactory performance, attributed to different ways of inducing defects. It is concluded that the deployment and validation of the proposed approach requires further testing that reproduces bearing degradation in automobiles.

### 3.3. AI Taxonomy Based on the Perceived Quality Framework (PQF)

Most car customers share their opinion about vehicle quality based on the combination of design, features, and previous experiences they have had with cars. Quantification of client needs can be achieved by perceived quality (PQ) during the phases of vehicle product development. PQ attributes are the characteristics that transmit the social, emotional, and functional advantages to the consumer. The PQ can be defined as the moment in which product, structure, and sensory elements interact with human experiences. A car assembly can work with 20–120 PQ attributes, which are responsible for the condition that sets the customer’s thoughts regarding the quality of the vehicle. The elaboration of an actual car is extremely complex and is impacted by PQ [67].

Quality perception is made by material and mental inputs, normally triggered by a tangible signal, which is processed through our senses, which are the basis of customer experience. Figure 5 describes the main human feelings that contribute to the initial position of the attributes of PQ, which are the following: visual quality, tactile quality, auditory quality, and olfactory quality.

#### 3.3.1. AI Taxonomy Based on Visual Quality

We found a document called “An Artificial Intelligence Platform Proposal for Paint Structure Quality Prediction within the Industry 4.0 Concept”. In this journal article, an artificial intelligence (AI) framework is proposed to create a quality forecast using big data. Car painting activities are complex. They require a large quantity of data, and the process is heterogeneous. The developed AI platform confirms the accuracy and validity of the initial phases. This works integrates an architecture to collect data with the big data solution, an analysis based on principal component analysis to reduce dimensionality. The quality of the painting process is assessed using specific spectrometers and measuring features. The proposed solution to the AI platform checks the availability of operating the neural network to foresee the quality of the procedure configurations [68].The next manuscript is called “A Case Study on First Time Quality Feature Investigation for an Automotive Paint Shop”. Considering automated vehicle body shops, manufacturing devices are typically implemented to coat automotive bodies in white. During the painting process, data is tracked, collected from sensors that control the painting in a consolidated process called the painted surface performance management (PSPM) method.In this research, machine learning (ML) approaches choose the main characteristics that could affect the result of the painting process, which needs further validation and research. The chosen configurations are used to set first time quality (FTQ) targets to forecast FTQ results. The study combined additional information to understand the relationship between the characteristics of machine learning and FTQ. ML was found to be a helpful tool for identifying the main characteristics that impact the target variables [69].The next investigation is “A perspective on the artificial intelligence’s transformative role in advancing diffractive optics”. AI methods such as machine learning and generative models enable the analysis of vast data sets to improve the accuracy of diffractive optical elements (DOEs), which are designed for certain tasks and requirements. The use of artificial intelligence approaches triggers the development of optical structures that handle light with high accuracy. In addition, AI improves production processes that result in better quality and productivity. In addition, AI methods speed up design iterations and prototyping. This consolidation of AI into several optics enables a great opportunity to change the functions of optical technology among different industries [70].

#### 3.3.2. AI Taxonomy Based on Tactile Quality

The next document is “Minimization of defects generation in laser welding process of steel alloy for automotive application” [71]. The text describes machine learning (ML) implementation to reduce and predict issue generation in laser welding processes for automotive uses. The result was a predictive quality control model with a prediction capacity precision of 61.4%. The principal results in terms of probability were that the generation of a welding depth defect is irrelevant, the probability of causing an internal problem is 15%, and, within optimized conditions, the probability of defects coming from false negative prediction is 15%.The next research is “Recent advances in artificial intelligence boosting materials design for electrochemical energy storage”. In the context of electrochemical energy storage (EES), artificial intelligence has a key role in innovation as the lead designer and inventor of batteries, fuel cells, and other materials. This study reviews machine learning tools in the developments of materials science, focusing on energy retention of battery technologies and fuel cell performance. In this paper, a comprehensive review is shared on AIs that have improved the study of advanced materials for EES, and therefore offer opportunities to discover new frameworks in materials science. Machine learning and deep learning can analyze high-volume data sets to identify trends and relationships. Furthermore, AI can improve synthesis and processing features, increasing the efficiency of the material development cycle. We found that the quality of the data is key and has an effect on the results of the models [72].

#### 3.3.3. AI Taxonomy Based on Auditory Quality

The initial article of this category is “Faulty voice diagnosis of automotive gearbox based on acoustic feature extraction and classification technique” [73]. The paper addresses the establishment of a systemic approach to diagnosing defects in automobile gearboxes through the analysis of defective sounds and ML techniques. The stages of continuous learning algorithms are described, including feature extraction, selection, and categorization. Five classification algorithms are used to model the gearbox fault diagnosis problem as an ML challenge. The main objective is to extract useful information from the sound signals to detect any issues in the gearbox component.The second article is “Auditory experience in vehicles: A systematic review and future research directions”. The purpose of this article is to investigate manuscripts on auditory experiences within a vehicle context and to share opinions about future opportunities to investigate this topic. The information collected was categorized according to the type of engine, and a contrastive analysis of the investigation tendency followed. The study demonstrated the existence of research topics in internal combustion vehicle (ICVs) not addressed in electric vehicles (EVs) and other growing research topics in the EV sector. Auditory sensations in vehicles are one of the elements of emerging driving experiences that can significantly improve customer satisfaction. As automated driving technology increases steadily, the continuous interconnection between people and vehicles is estimated to increase; therefore, investigation of auditory customer interfaces is the key to improving the quality of the driving experience [74].

#### 3.3.4. AI Taxonomy Based on Olfactory Quality

The first article of this subcategory is “Self-validating sensor technology and its application in artificial olfaction: A review”. A sensor is equipment that detects changes in concrete quantities and transforms them into electrical energy. Automated detection equipment depends on precise and correct sensor measurements. The trustworthiness and maintainability of the system can be improved with an intelligent functional structure to track sensor status delivered by self-validating sensor technology. The consistency and robustness of gas chemical sensors for artificial olfaction have limited the mass production of the electronic nose system. In this paper, self-validating sensor technology applications were systematically reviewed, with briefings on the deployment procedures, demonstrating its capability in artificial olfaction [75].

We created a word hub of the summaries described in Section 3, using a word cloud tool to identify which were the main repeated AI tools. The results show that those principal tools are PCA, neural networks, machine learning, and artificial intelligence. Other general concepts that are more recurrent are model, data, manufacturing, quality, fault, automotive, systems, and detection. These concepts are visualized in Figure 6. In Section 4, we will share our thoughts about the results of our review of the literature.

## 4. Discussion

The deployment of artificial intelligence (AI) in the automotive industry has demonstrated a key impact on quality management, basically detecting problems, estimating failures, optimizing processes, and improving operational efficiency. The categorization of AI applications in the automotive industry, organized into the proposed taxonomy (Figure 2) of quality management (QM), zero-defect manufacturing (ZDM), and perceived quality framework (PQF), enables a detailed review of the benefits and limitations of these technologies in the sector.

Regarding AI’s effectiveness in the automotive sector, the reviewed manuscripts emphasize that AI tools such as artificial neural networks (ANN), deep learning (DL), and principal component analysis (PCA) have importantly upgraded prediction and issue-identification capabilities with high precision. In the case of quality control, machine-learning-based approaches have enabled on-time part inspection, decreasing dependence on manual inspections, and improving early defect detection in highly automated manufacturing lines.

In addition, in zero-defect manufacturing (ZDM), AI has enabled the projection and prevention of errors in key systems such as internal combustion engines, drum brakes, and electronic components. However, some studies have shown that the effectiveness of these solutions has great dependence on data quality and consolidation with other intelligent production systems. Concerning the fundamental benefits of AI in quality management in the automobile industry, we note that the automation of quality inspection decreases the requirement for a top-qualified workforce and reduces manual errors, enhances manufacturing planning and waste diminution with digital solutions such as deep learning (DL) and neural networks, and improves defect estimation capabilities in critical parts. In addition, the developed predictive methods have shown advances in inventory management efficiency and spare parts demand projection.

Although its benefits are numerous, AI also faces several challenges and limitations in its deployment in the automotive industry in the following ways: The capability of AI models is directly linked to the quantity and quality of accessible training data, which means that wrong data can generate inaccurate forecasts and deficient decisions. Certain AI methods have faced challenges in their interpretation, which can lead to low popularity in strict production sites. Moreover, upgrading production activities with AI requires important investment and adaptation in training and infrastructure. Finally, even though several solutions have demonstrated exciting results in validation scenarios, scaling them to high-volume applications remains a challenge.

AI deployments have been particularly useful in solving specific problems in the automotive industry, including identification of surface defects in auto parts using computer vision models, estimation of errors in engines and transmission systems using predictive analytics tools, on-time follow-up of production processes, improving efficiency and eliminating waste, optimization of preventive maintenance, and decreasing breakdown periods and repair costs.

To maximize the positive impacts of AI in quality management in the automotive industry, the following strategies should be considered: deployment of data management plans that ensure precise and consistent information, promotion of the usage of AI models to enable their approval in the sector, development of adaptable and expandable approaches that can be integrated with actual technologies without demanding high infrastructure upgrades, and boosting the consolidation of expert knowledge with AI systems to improve decision making.

We developed Table 1, which include the name of the article, the category and subcategory they belong to in our taxonomy, and the AI application that was included in the manuscript, as well as the solution that each AI supports and the limitations each of them faces.

In Section 5, we will share our main findings and trends regarding AI applications in the automotive industry in the future.

To increase the relevance of our literature review, we developed Table 2, which include relevant industrial reports on the application of AI quality control in the automotive field, with key elements to enrich the information for researchers and professionals in the automotive industry.

## 5. Conclusions

The deployment of AI within the automotive sector has had an important effect on quality management, with progress in the identification of problems, the estimation of failures, the improvement of processes, and the enhancement of operational efficiency. This systematic review has determined how several AI tools, such as deep learning, artificial neural networks (ANN), and principal component analysis (PCA), have changed quality control activities in automotive production.

One of the main findings of this literature review is that AI has helped automate essential tasks in production, decrease the dependance on human inspections, and improve early defect identification. AI solutions have demonstrated particular effectiveness in the inspection of live vehicle components, enabling organizations to reduce waste and increase the reliability of products. Furthermore, AI has driven the development of zero-defect manufacturing (ZDM) plans, with special attention paid to estimating and avoiding claims before they have an effect on manufacturing.

AI has launched fundamental innovations in quality management in the automotive industry such as on-time manufacturing tracking through sensors and machine learning algorithms, enabling proactive problem detection, enhancement of preventive maintenance by forecasting equipment shutdown before it happens, quality control procedures following computer vision utilizing convolutional neural networks (CNN) to inspect automotive components with great precision, and smart supply chain management. This progress has not only improved operational efficiency but also improved the industry’s sustainability by decreasing material waste and environmental impact.

Future research should pay special attention to the following elements: the development of transparent AI methods to improve the comprehension of the algorithms that were used, the combination of AI tech tools with existing production models, ensuring a steady transformation to Industry 4.0 and 5.0, and the evaluation of the effect of AI on the security and reliability of auto parts to ensure that the solutions meet more strict regulatory standards.

The future of AI in automotive quality is leading to greater consolidation with cyber-physical systems and the application of digital twins for process simulation and optimization. Other important trends are the extension of computer vision solutions for live issue identification with higher precision; AI deployment in the design and production of advanced materials that improves the constitution of metal combinations and structural components; the usage of generative models for issue estimation and correction before manufacturing, which decreases costs and lead time; and finally, increase the deployment of AI in the area of perceived quality to foster a better customer experience and ensure customer satisfaction.

AI tools in automotive R&D, such as copilots for the fulfillment of international standards like ISO and simulations supported by AI, have enhanced engineering processes, software validation, and product creation. Real-world examples of AI applications, like the combination of ChatGPT with Mercedes-Benz vehicles for user interaction and R&D assistance, confirm AI´s ability to optimize processes and customize services while handling cybersecurity and data privacy risks. In spite of its positive impacts, scalability issues are still visible, mainly due to cultural paradigms, infrastructure requirements, and regulatory barriers. AI´s success depends on complete data governance, clear processes, and adaptation capacity, as found in the industrial reports of our review. Future plans should consider the combination of AI with connected systems, improved data management platforms, and fulfillment of global standards to meet evolving regulations and market requirements.

In summary, AI is becoming a primary tool for quality improvement in the automotive sector, enabling more efficient, accurate, and sustainable processes. As technology evolves rapidly, AI solutions are expected to expand and improve, overcoming actual challenges that set the pace for innovation in vehicle production. The key to success in this area is to develop robust, understandable AI methods that smoothly consolidate with actual technologies, ensuring a positive and viable impact on the industry.

## Figures and Tables

**Figure 1 sensors-25-01288-f001:**
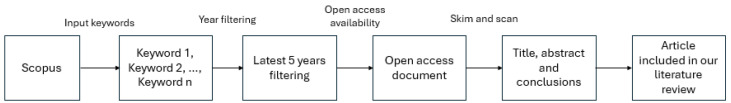
Literature review research on Scopus.

**Figure 2 sensors-25-01288-f002:**
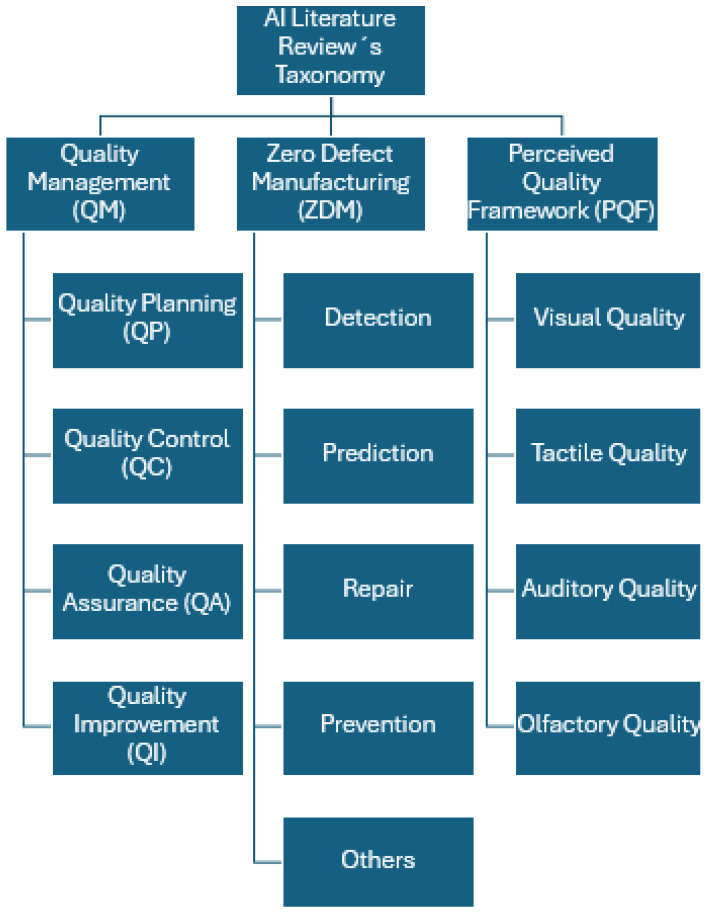
Proposed taxonomy of AI applications.

**Figure 4 sensors-25-01288-f004:**
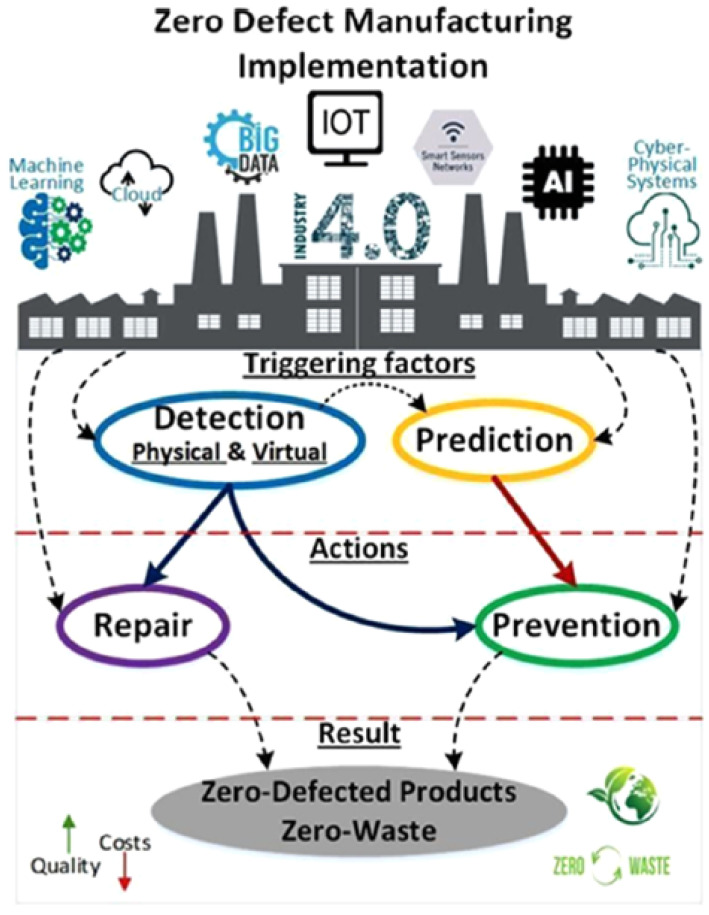
Framework for ZDM [10].

**Figure 5 sensors-25-01288-f005:**
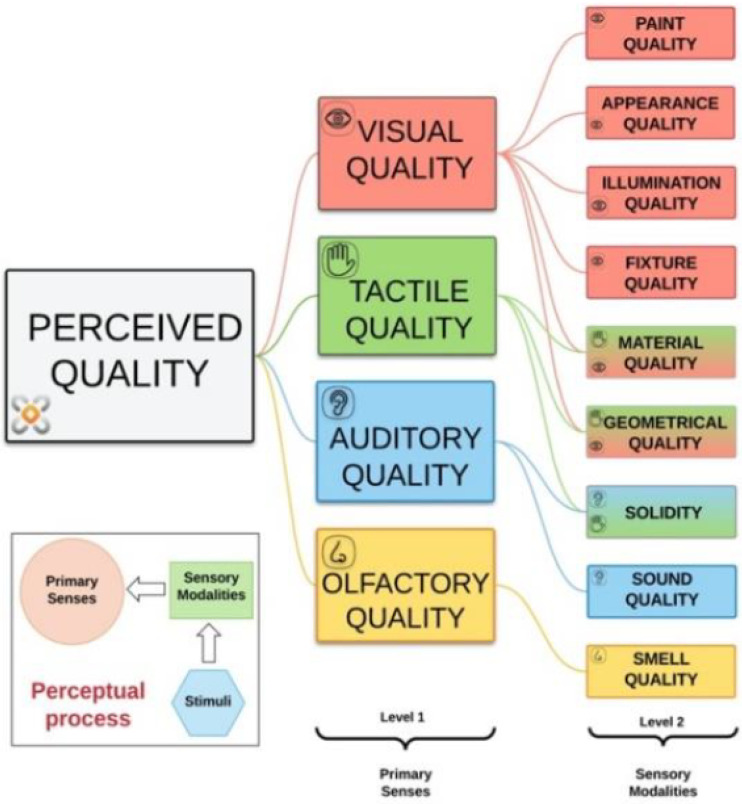
PQF (perceived quality famework) according to [67].

**Figure 6 sensors-25-01288-f006:**
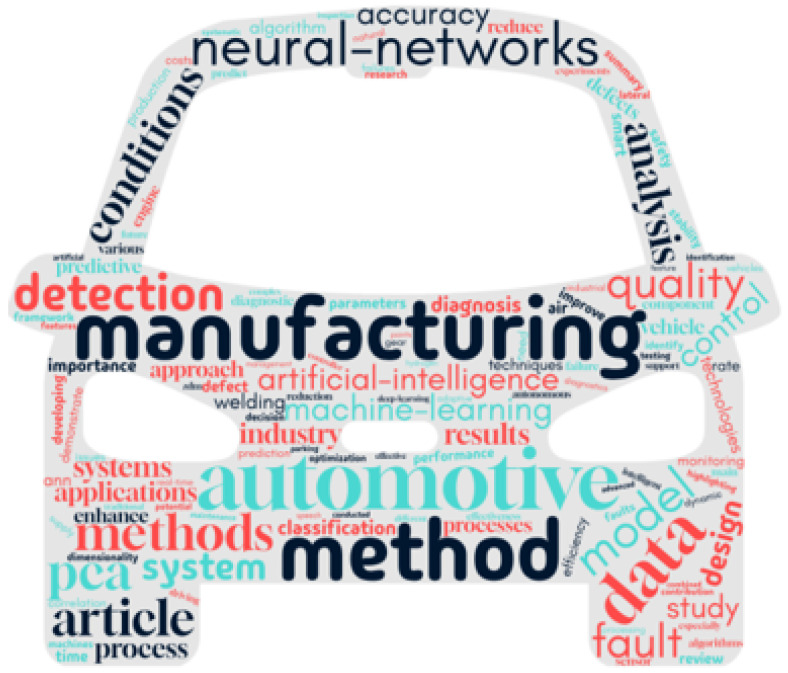
Word hub of the summaries of the literature review.

**Table 1 sensors-25-01288-t001:** Taxonomy of artificial intelligence applications.

Article	Taxonomy: Category/Subcategory	AI Application or Technique	Solution/Support	Limitations (Discussions and Conclusions)
[7]	ZDM/Repair	Artificial intelligence (AI) and related tools	Review of AI’s benefits for automotive industry, i.e., vehicles are more intelligent, safer, reliable, automated with increased efficiency.	Lack of real scenarios for AI implementations to enhance the understanding of the applications.
[8]	ZDM/Prediction	Artificial neural networks (ANN), evolutionary algorithms	Reduction of production deterioration by 10% and manufacturing time by 62%.	The method relies on specific raw materials and production activities, making it difficult to apply to other sectors.
[10]	QM	Industry 4.0/Quality 4.0 technologies, robotics, IoT	The document indicates the importance of QM deployment using technologies within Industry 4.0 and Quality 4.0 frameworks, considering the operational and finance impacts.	The article does not provide specific case studies or examples of Quality 4.0 deployments in real scenarios.
[11]	QM/QP	Machine learning (ML), deep-reinforcement learning	AI improves forecasting, capacity planning, and order release. ML enhances prediction accuracy.	Problems arise from difficulties in handling orders, control of supply and demand, and ineffective order approval processes.
[12]	QM/QP	Regression models, artificial neural networks (ANN)	AI techniques like regression and ANN improve demand forecasting accuracy for spare parts.	No limitations are specified; only obstacles related to spare part existence and cost are included.
[13]	QM/QP	Knowledge-based systems (KBS), machine learning (ML)	KBS and ML help optimize scheduling and maximize production efficiency in electric drive manufacturing.	No specific limitations mentioned.
[14]	QM/QP	Convolutional neural networks (CNN), machine learning (ML)	CNN and ML are utilized to forecast early quality within the manufacturing process using sensor data, improving the categorization accuracy.	While no explicit limitations were identified, potential issues include real-time data processing and a potential increase in the complexity of sensor data.
[15]	QM/QC	Random forest (RF), gradient boosting trees (GBT), fully connected neural networks (FCNN)	Usage of preventive models to improve reliability and quality of metallic additive manufacturing (MAM).	Machine learning models lack transparency, making their decision-making processes difficult to understand.
[16]	QM/QC	Convolutional neural networks (CNN), machine learning (ML)	Projection of product quality through sensor data and corrective action implementation.	No explicit limitations are stated; however, potential challenges may include sensor data inconsistencies and real-time adjustment difficulties.
[17]	QM/QC	Supervised machine learning (ML)	Projection of final product quality, enhancing inspection efficiency, and making cost savings.	No explicit limitations are mentioned, though difficulties might arise in data integration and infrastructure deployment.
[18]	QM/QA	Artificial intelligence (AI) and related tools	Automation of monitoring, which positively impacts on delay decrease and human workforce dependency.	No specific constraints are identified, but AI integration and the complexity of new technologies could present challenges.
[19]	QM/QA	Deep learning (DL) for defect detection	Identification of physical issues with a high rate of precision.	The model’s effectiveness is limited to surface defect detection and may not be generalizable to other product types.
[20]	QM/QA	AI-based supervision controller	Reduction of deviations among the required and actual surface roughness, increasing product quality.	No explicit limitations are highlighted, but obstacles in execution and machine compatibility may occur.
[21]	QM/QI	AI techniques (Bayesian networks, fuzzy logic, ANFIS)	Strengthening of machining process and enhancement of quality, efficiency, and sustainability.	Availability of raw material availability and cost increase as part of AI integration.
[22]	QM/QI	IoT, AI, Quality 4.0 concepts	Enrichment of productivity, product design, and customer experience.	No explicit restrictions are mentioned; however, scaling and integrating these technologies across diverse manufacturing environments could be difficult.
[23]	ZDM/Detection	Artificial intelligence (AI), machine learning (ML)	Increase of quality management and handling of complex quality defects.	Inappropriate data and wrong organization of released products decrease model precision. Additionally, the time required to establish ZDM architecture was a significant barrier.
[24]	ZDM/Detection	Artificial intelligence of things (AIoT)	Fulfillment of 100% accuracy in error identification of rotating machines.	No specific limitations or challenges are addressed, though additional testing could be beneficial.
[25]	ZDM/Detection	Artificial intelligence (AI)	Fault identification with a precision ranging from 80% to 100%,.	A potential constraint is the reliance on simulated data, which may not represent real-world scenarios.
[26]	ZDM/Detection	Artificial neural networks (ANN)	Effective identification of typical claims in automobile drum brakes with high precision.	Limitations may involve further validation in practical scenarios.
[27]	ZDM/Detection	Machine learning (ML) models	Integration of operators, and forecasting of worker fatigue fostering better working conditions.	Fatigue estimation accuracy could be impacted by various uncontrolled factors.
[28]	ZDM/Detection	Semi-supervised learning (SSL)	Literature review that covers error identification and diagnosis in industrial environments, highlighting future investigation needs.	Difficulties in results comparison among setting and deployments of SSL.
[29]	ZDM/Detection	Machine learning, PCA, random forest (RF)	Dimensionality reduction and enhancement of detection precision in automobile HVAC system fault diagnosis.	The study does not detail limitations, but additional testing could assess how the model reacts across different HVAC systems or contexts.
[30]	ZDM/Detection	Deep learning, PCA	A revision of AI techniques and CCD camera-based systems used in on-time laser welding tracking and quality evaluation.	There is limited discussion of the practical challenges of applying these AI techniques in industrial contexts, along with the need for continued development in smart quality evaluation systems.
[31]	ZDM/Detection	PCA, artificial neural networks (ANN)	Creation of an approach for the automatic identification and categorization of welding incidents.	The article lacks a thorough discussion of real-world situations and issues such as data quality, expandability, and the consolidation of this approach into actual industrial workflows.
[32]	ZDM/Detection	PCA, backpropagation neural network (BP)	Model used to estimate the sizes of issues in materials with prediction error of 4% to 10%.	The study does not extensively explore difficulties related to the transferability of the model or its results under changing material conditions and real-world adoptions.
[33]	ZDM/Detection	Principal component analysis (PCA)	Production of top-quality images of mistakes that enhances lateral resolution and reduces the average defect sizing error by 48.81%.	It remains unclear how well the method performs in non-ideal conditions or in manufacturing environments with fluctuating noise and interference levels.
[34]	ZDM/Detection	Principal component analysis (PCA), decision trees	Diagnosing multiple early faults in induction motors at several frequencies	Performance may be affected due to noise or incomplete data, particularly in real-world applications with changing operating conditions.
[35]	ZDM/Detection	PCA, K-means, support vector machine (SVM), SSA-PCA-SVM	Dimensionality reduction and improvement of diagnostic precision to approximately 90%.	Real-world applications may have limitations where environmental variables or unexpected leak patterns affect precision.
[36]	ZDM/Detection	PCA, atificial neural networks (ANN)	Kinematic chains organization with a 90.8% reliability	As the approach fulfills high precision, effectiveness may change according the several fault types or operating conditions.
[37]	ZDM/Detection	PCA, supervised and unsupervised classifiers	Innovative diagnosis for automobile fuel cell issues, reducing dimensionality with PCA.	Consistency and reliability across diverse processes remains a challenge. Additional validation is required.
[38]	ZDM/Detection	Radial basis function neural network (RBFNN)	Progress in curvature smoothness, security, and parking efficiency are shown.	Real-world verification and further validation in multiple parking situations are needed for broader applicability.
[39]	ZDM/Detection	Fisher discriminant analysis (FDA)	Model for detecting issues in automotive engine test beds.	Deeper research is key to checking the adaptability to more complex systems and a broader range of error types.
[40]	ZDM/Detection	Hidden Markov model (HMM)	Incident identification in automotive engines, reaching a 4.33% false alarm rate and a 24.83% missed detection rate.	Conventional statistical models are not suitable for engines, even though the proposed method still has a potentially high missed detection rate.
[41]	ZDM/Detection	Artificial intelligence and related tools	Systematic review of defect types and diagnostic strategies for electrified drive powertrain systems (EPDS).	Due to unknown failure modes, it is essential to foster innovative assessment models.
[42]	ZDM/Detection	Model-based fault diagnosis	Development of a model to detect undiagnosed problems of automotive air intake system of combustion motors.	Future investigation will emphasize the expansion of a hybrid engine model and designing virtual sensors.
[43]	ZDM/Prediction	Artificial intelligence and its related tools	Identification of AI in automotive production, emphasizing its potential to minimize functional costs and boost efficiency.	Limitations in generating quantitative results that achieve scientific standards.
[44]	ZDM/Prediction	Artificial neural networks (ANN), random forest (RF)	AI model for online quality inspection to estimate load failure and weld quality, fulfilling superior precision.	Limitations are not described, though improvements in ANN method accuracy are suggested.
[45]	ZDM/Prediction	Neural networks (Bayesian regularization)	Robust manufacturing quality through simulation.	The research does not explicitly discuss relevance and adaptability limitations, but simulation results indicate potential outcomes.
[46]	ZDM/Prediction	Fuzzy logic, neural networks, machine learning	Automotive AI deployment to enhance corporate knowledge management and decrease warranty costs.	Limitations are not discussed, but recommendations that broader AI applications could enhance customer satisfaction and product quality across various automotive domains are shared.
[47]	ZDM/Prediction	Artificial intelligence and related tools	Fast precision issue recognition, optimizing tasks in smart factories.	No limitations are discussed, but AI impacts are highlighted.
[48]	ZDM/Prediction	Machine learning (ML)	Machine learning’s role in predictive maintenance (RdM) for the automotive industry is emphasized, specifically in drivetrain systems, indicating cost savings and enhanced prediction capability.	Challenges in implementing predictive maintenance (RdM) are identified, basically related to functional safety while keeping cost efficiency.
[49]	ZDM/Prediction	Machine learning (ML), principal component analysis (PCA)	Projection of automotive´s welded joints quality using machine learning and PCA.	The manuscript does not thoroughly examine the practical difficulties of adapting this approach to large-scale manufacturing facilities.
[50]	ZDM/Prediction	Machine learning (ML), Q-learning, decision trees, CNNs	AI defect categorization, and supervised/unsupervised approaches for predictive maintenance.	The research does not assess flexibility or adaptability issues across industries.
[51]	ZDM/Prediction	Deep learning, recurrent neural networks (RNNs)	Review of speech recognition publications to discuss updates in feature extraction and hybrid approaches for improved identification precision.	More research about new feature extraction is recommended.
[52]	ZDM/Prediction	Hybrid deep learning, feature selection	Projection of errors in automotive manufacturing. with high conformance.	No constraints are expressed, but future advancements could enable wider industrial solutions and on-time data incorporation.
[53]	ZDM/Prediction	Deep learning	Forecast of automotive production quality control tests, enhancing efficiency by 15%.	Fulfilling a balance between the negative prediction rate (NPR) and the false emission rate (FOR) remains a key condition.
[54]	ZDM/Prediction	Adaptive neural network (ANN)	Elevating the balance and trajectory tracking precision, especially under difficult driving scenarios.	Further refinement in different environmental and operational contexts is necessary.
[55]	ZDM/Prediction	Principal component analysis (PCA)	Increment casting quality by targeting issue reduction.	Additional investigation is needed to optimize other process characteristics.
[56]	ZDM/Prediction	Principal component analysis (PCA)	Projection of failure time of copper wire bonds, which is key for forecasting the reliability of parts.	Improvement of accuracy depends on data volume, yet dependence on data quality remains an issue. Additional studies are needed to confirm transferability across several products and environments.
[57]	ZDM/Prediction	Adaptive neuro-fuzzy inference system (ANFIS)	A model is made to estimate the thermal performance of a natural convection heat transfer system by sharing accurate results.	The developed model demonstrated high forecasting accuracy, but adaptability to more complex methods or different product types remains unexplored, and its resilience for real-time adaptations or diverse sustainable conditions has not been assessed.
[58]	ZDM/Repair	Artificial intelligence for optimization in AM (additive manufacturing)	Systemic review of AI applications for improving repair and restoration in metal additive manufacturing (AM). It facilitates repair processes through AM methods to enhance part design.	AI practices in AM are limited, requiring systematic approaches for further development. In addition, it does not review challenges such as material constraints and industrial adaptability.
[59]	ZDM/Repair	Advanced AI, IoT, digital twins	Analysis of the progression of industrial revolutions, highlighting Industry 5.0, which emphasizes human–machine partnership.	Lack of detailed examples or real-world adoptions of these technologies. Moreover, it does not thoroughly review the consolidation of challenges within actual production methods or workforce adaptability requirements.
[60]	ZDM/Prevention	Artificial intelligence and its related applications	Consolidation of prediction and inspection of automotive coating quality control evaluation and decreasing defects of the vehicle topcoat application.	Absence of specific challenges encountered during the method adoption or sharing of benchmarks with other industry practices.
[61]	ZDM/Others	Automated model creation, interpretable AI models	Demonstrating the model’s efficiency in automatically creating interpretable approaches for embedded control applications while fulfilling expected results.	Omission of detailed discussion of its limitations or potential challenges in applying automation to more complicated or large-volume control systems. Further research on adaptability and robustness is suggested.
[62]	ZDM/Others	AI, intelligent robots	Systemic review of AI consolidation in Indian production, highlighting the need for better infrastructure and policies to facilitate AI adoption. The importance of updating educational models to align with technological advancements is also remarked upon.	Limitations are noted in terms of lack of infrastructure, government policies, unemployment concerns, and skills gaps, without proposing specific solutions.
[63]	ZDM/Others	Fisher discriminant analysis (FDA), principal component analysis (PCA)	Defect diagnosis in vehicle gearboxes with FDA/PCA approaches, with resulting FDA showing higher precision and less monitoring costs compared to PCA. Average recognition is enhanced by approximately 14%.	The study is focused on the gearbox only; therefore, it does not explore adaptability to other components.
[64]	ZDM/Others	Principal component analysis (PCA)	The article identified the key process parameters of the studied electrical discharging machining (EDM) process, so-called Lp and Ton parameters, which are typically related to quality.	Lack of detection of interaction impact with other features of Lp and Ton, as critical parameters.
[65]	ZDM/Others	IoT	Enhancement of precision and reliability of autonomous vehicle parking. It maintains stable radiation patterns across frequencies, which is crucial for RTLS systems.	There is a gap for further validation and testing needed for real-world scenarios in autonomous vehicles.
[66]	ZDM/Others	K-nearest neighbors (KNN), binary differential evolution	Early degradation identification and error categorizations of automotive bearings with an effective rate of 79.78%.	More testing with bearing degradation simulations in automotive applications is necessary for verification and adoption.
[68]	PQF/Visual Quality	AI, big data, principal component analysis (PCA), neural networks	Projection of paint quality in the automotive sector, which is evaluated through specific measuring features.	The complexity of the painting process and the high data volume may delay or make harder the AI platform’s adaptability and real-time implementation. Additionally, consolidating diverse data and ensuring neural network estimation precision under variable conditions needs further validation.
[69]	PQF/Visual Quality	Machine learning (ML)	Detection of critical characteristics negatively impacting automotive painting; support for FTQ target setting and forecast results.	Needs more validation and investigation to confirm the chosen features. Relationship between ML and FTQ requires further research.
[70]	PQF/Visual Quality	Machine learning (ML) and generative models	Improving the design and precision of DOEs, optical structures, their quality and productivity.	Needs more validation and investigation to confirm the chosen features. Relationship between ML and FTQ requires further research.
[71]	PQF/Tactile Quality	Machine learning, predictive modeling	Development of a predictive quality control model using machine learning to reduce defects of laser welding in automotive components with an accuracy of 61.4% and a probability defect identification of 15%.	There is a gap in terms of improving the accuracy rate (61.4%). In addition, there are challenges caused by false negative projections related to defect production. Finally, model optimization may be necessary to improve estimation reliability and reduce false negatives.
[73]	PQF/Auditory Quality	Machine learning, classification algorithms	Evaluation and classification of gearbox defects by analyzing sound signals using machine learning.	Uncontrolled factors could impact the transparency and quality of sound data. Fault detection precision may be limited by the feature extraction process, which might overlook subtle defect characteristics. Furthermore, fine-tuning multiple classification algorithms is essential to achieving optimal results.
[72]	PQF/Tactile Quality	Machine learning and deep learning	Enhancing battery and fuel cell development. In addition, detecting trends and relations in vast datasets.	The data quality considerably impacts method precision and outputs. More investigation is required to boost AI implementations in material science.
[74]	PQF/Auditory Quality	AI in auditory experience analysis	Classifying and analyzing investigation trends of auditory experience in ICVs and EVs.	More investigation is required to improve auditory customer interfaces.
[75]	PQF/Olfactory Quality	IoT	Enhancing the reliability and maintainability of automated sensor systems with deployment of artificial olfaction.	Further investigation is needed for large-scale implementation of electronic nose systems.

**Table 2 sensors-25-01288-t002:** Industrial reports of AI applications in the automotive sector.

Report	Ethical/Regulatory/Cybersecurity Risk/Data Privacy Risk	Scalability Obstacles	Comparative Analysis of AI Models	Real-World Case	Defect Rate Comparison Before vs. After IA	AI Adherence to Standards/Certifications
[76]	The study shares concerns about data privacy and regulatory risks when deploying AI in automotive R&D. It emphasizes challenges related to cybersecurity and fulfillment with data protection regulations. Solutions like transparent terms of service for AI agreements are given.	Obstacles of AI expandability are related to strong computational infrastructure, asset management, flexibility for technological evolution, and a modular data platform.	This information is not explicitly included.	Core AI use cases are in engineering, software verification, homologation, and product design.	This information is not explicitly included.	AI methods will be used as copilot applications to support employees on manual tasks, for example writing procedures for fulfilling ISO standards.
[77]	This information is not addressed.	Cultural and technological features are the main elements of scalability issues that can be solved with best practices deployment and examples of success for interested parties.	Typical techniques like computational fluid dynamics and finite element analysis are still predominant for simulation purposes over ML and AI, which are not fully consolidated in product development activities.	Simulation product performance like cloud-based platforms, closed loop simulation, and digital twin tools.	This information is not addressed.	This information is not addressed.
[78]	Worries regarding data privacy are described when combining customer physical safety and digital safety. Data is stored in Mercedes´ Microsoft Cloud and is not sent back to Open AI or ChatGPT-4. A risk for ChatGPT is the large language model on which it is built. Ethical concerns, like the handling of structured and unstructured data, are also highlighted. Another risk is the connections among internal and external systems. Solutions like setting transparent guidelines and disclaimers are provided.	Challenges relate to handling transitions in terms of education and culture. Other factors such as creativity, human interaction, and leadership are important to keep.	ChatGPT is compared with typical voice assistants in terms of conversational skills and continuous learning.	Gen AI: Power voice assistants to enhance software; sales and aftersales offering call summarization: personalized marketing delivering customized campaigns; diagnosis and repair to rapidly identify and solve an issue; and R&D by linking CAD drawings, protocols, and emails. Mercedes-Benz ChatGPT beta program: shares navigation and car control features implemented in ChatGPT.	This information is not addressed.	This information is not addressed.
[79]	Argues ethical issues, regulatory difficulties, cybersecurity risks, and data privacy problems related to organizations are steadily focusing on responsible AI, considering data governance, process standardization, automation, and protocol establishment. In addition, AI top performers have identified risks related to privacy and impartiality.	Mentions that AI leaders are more likely to implement best practices for the industrialization of AI, which would led to narrowing and shortening the opportunities for AI low-performers.	This information is not addressed	Sustainability, environmental impact and waste reduction, energy efficiency, enhancement of service processes, product/service development, marketing and sales, risk management.	Describes enhancements in AI-driven defect detection but does not shares a direct before-and-after comparison.	This information is not addressed.
[80]	The report shares several challenges in terms of cyber security and liability. Other implications in the study are considered, like software reliability, system failure potential, and rapidly changing nature of driving behavior.	Difficulties in scalability are described, basically referring to regulatory resistance, technical issues, and doubts about implementing new solutions.	Certain elements linked to EV (electric vehicles), like maintenance demand, repair, and battery management, need a revolution in traditional insurance methods and a change to a new approach focused on software-driven and connected solutions.	AI and ML in road safety. Moreover, the text demonstrate how insurers are utilizing AI to optimize complaint handling and fraud identification, enabling operational efficiencies and cost reductions.	This information is not addressed.	The text mentions that OEMs request that key repair and maintenance must be performed by certified technicians to keep warranty coverage.
[81]	Boards should make sure that leaders and their employees set a high standard for cybersecurity, ensuring that security is embedded by design in digital products and that tech teams own awareness for cybersecurity.	This information is not explicitly included.	This information is not explicitly included.	For industrial settings, space machine learning automatically enhances AI-enabled systems, helping managers and directors to maintain the rhythm of worldwide processes.	This information is not explicitly included.	This information is not explicitly included.

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
