# Peer review of "Artificial Intelligence for Quality Defects in the Automotive Industry: A Systemic Review"

_sensors, 2025, doi:10.3390/s25051288_

Round 1
Reviewer 1 Report
Comments and Suggestions for Authors
Dear authors,
The abstract should be reworked a bit, I think it should be more focused and better emphasized the theme of the abstract.
Another suggestion would be perhaps the conclusion can be extended a little bit and keywords perhaps it would be better to be in alphabetical order.
The title would perhaps be shorter and more resonant.
Many of the pictures have visibility problems, are blurry, and should be redone for example: fig.3, fig. 4.
On page 5 at the top there are some editorial mistakes in the first paragraph, the article should be revised, perhaps item 3.2 a little more detailed in explanation.
Paragraph 3.2.4 should be re-worded entirely.
The similarity percentage is high, please review areas with high similarity, it is not recommended especially in the conclusions part to have similarity. First the similarity issue needs to be reviewed and then the article reviewed again.
Many subheadings are on the last line of the page, and it is not editorially correct, on page 13 is item 3.3.3 which is at the bottom of the pages.
As a conclusion the paper should be reworded in many places and carefully re-read to have a logical thread, some paragraphs need rearranging to make the paper fluent. After rereading and reworking the paper it can be submitted again for review.
Best regards,

-
Author Response
Please see the attached pdf file.

Reviewer 2 Report
Comments and Suggestions for Authors
The manuscript presents a comprehensive review of the current state and potential applications of AI in addressing quality issues within the automotive industry. However, there are several significant shortcomings that necessitate rejection. Here are the detailed reasons for this decision:
1. The taxonomy proposed in Section 3.1 is too general and lacks specificity. Terms like "Quality Planning," "Quality Control," "Quality Assurance," and "Quality Improvement" are not sufficiently refined to differentiate between various AI applications and their impacts on quality management.
2. Many sections of the paper provide only a cursory overview of the literature without delving into the methodologies, findings, and implications in depth. For instance, the discussion on machine learning and neural networks in section 4 is too general and does not critically analyze the effectiveness or limitations of these technologies in real-world applications.
3. There is no systematic comparison or evaluation of the different AI techniques discussed. Readers are left without a clear understanding of which methods are most effective, under what conditions, and why.
4. The paper’s organization is disjointed, with abrupt transitions between sections that make it difficult to follow the logical flow of ideas. For example, the shift from the introduction to the materials and methods section is abrupt and lacks a smooth transition.
5. The manuscript includes many redundant references and repetitive statements, which detract from the clarity and conciseness of the text.
6. While the paper aims to provide a systemic review, it fails to offer new insights or perspectives that significantly advance the field. Instead, it largely reiterates existing knowledge without contributing original research or novel analysis.
7. The conclusion section is too brief and does not adequately summarize the key findings or provide actionable recommendations for future research or application.
By addressing these issues, the manuscript could potentially be resubmitted after substantial revisions. However, as it stands, the current version does not meet the standards required for publication in this journal.
Author Response
Please see the attached pdf file.

Reviewer 3 Report
Comments and Suggestions for Authors
The article "Artificial Intelligence and Its Application to Solve Quality Defects in the Automotive Industry: A Systemic Review" by O.M. Matamoros and colleagues is devoted to the analysis of publications on the current state of research related to the use of artificial intelligence (AI) in quality control in the automotive industry.
The article is a review-type, and in general, the papers mentioned in the manuscript, are relevant to the topic. The article is written in competent technical language and has an acceptable volume.
I suppose that the article has imperfections both in terms of its structure and sections, which are discussed below.
Abstract. In general, the abstract looks acceptable, but it is recommended to exclude a reference to another publication, and to reduce the information and criteria for articles searching.
Introduction. The introduction contains some unnecessary information that is not related to the essence of the study itself. The information presented in lines 47-73 does not directly relate to the article topic, and was not used in the article further.
Materials and Methods. This section contains some unnecessary details of the papers search. Information about how many papers remained after filtering at each stage is uninteresting and unnecessary for readers. This section can be shortened to one paragraph without losing meaning.
Taxonomy. The format of presenting article data is quite controversial. This format of presentation, in which there is no detailed analysis of the presented information, but only some systematization of the found articles is presented, is quite difficult to read. The lack of graphic materials also does not contribute to the easy reading of this section.
Discussion. This section is too short and sparse. It is expected that if the authors did not conduct an in-depth analysis of the obtained information in the previous section, then this analysis will be carried out in “Discussion” section. However, there is no analysis of the information in this section.
Based on the manuscript title, it was expected that the paper under review would present the current level of achievements in the field of AI in automotive industry defect control. Particularly, which operations are already being perfectly solved using AI, which tasks can be solved in the very near future, which tasks cannot be solved today and tomorrow, and which tasks are irrational to solve. The same could be said about specific products or parts of an automobile - how rational and possible is the use of AI for specific units or parts.
It was expected to see a discussion on how the use of AI could be improved. Which circumstances and factors limit the 100% use of AI in quality control in this industry?
In addition, the specifics of the industry (automotive industry) are not sufficiently reflected. What are the features of AI using specifically in the automotive industry?
Also, the title of the article talks about "quality defects", but even after reading the article, it is difficult to understand what exactly the authors mean by “quality defects”.
Unfortunately, most of these questions are not answered in the article. I believe that the article should be revised in terms of the paper design, expansion of the “discussion” section and answers to the questions posed.
Author Response
Please see the attached pdf file.

Round 2
Reviewer 1 Report
Comments and Suggestions for Authors
Dear authors,
The paper has been reformulated according to the comments.
Now from my point of view it is much more logical and better emphasizes the subject.
But the percentage of similarity remained quite high, if the journal accepts such a percentage, I have nothing against it, in most cases the percentage of similarity should be below 15%.
Best regards,
Author Response
Please see the attached pdf file.

Reviewer 2 Report
Comments and Suggestions for Authors
The review article provides some valuable insights for the industry; however, it lacks sufficient references to support its claims. The article would benefit from a more comprehensive citation of relevant studies. Additionally, the logical structure of the paper is weak, and the flow of ideas could be improved for better clarity and coherence.
Author Response
Please see the attached pdf file.

Reviewer 3 Report
Comments and Suggestions for Authors
The article "Artificial Intelligence and Its Application to Solve Quality Defects in the Automotive Industry: A Systemic Review" by O.M. Matamoros and colleagues (revised version)
The authors have done sufficient changings in the article structure, excluding information that contained unnecessary details of the review methodology, and significantly expanding the "Discussion" section. Most of my comments were eliminated. The article now contains more information on the use of AI to solve quality control problems specifically in the automotive industry.
In my opinion, this revised version of the article is understandable and useful to the reader.
Author Response
Please see the attached pdf file.
